# Phenotypic and Molecular Characterization of Carbapenem-Heteroresistant *Bacteroides fragilis* Strains

**DOI:** 10.3390/antibiotics11050590

**Published:** 2022-04-27

**Authors:** Zain Baaity, Friederike D. von Loewenich, Elisabeth Nagy, László Orosz, Katalin Burián, Ferenc Somogyvári, József Sóki

**Affiliations:** 1Institute of Medical Microbiology, Albert Szent-Györgyi Health Centre and School of Medicine, University of Szeged, H-6725 Szeged, Hungary; baaity.zain@med.u-szeged.hu (Z.B.); nagy.erzsebet@med.u-szeged.hu (E.N.); orosz.laszlo@med.u-szeged.hu (L.O.); burian.katalin@med.u-szeged.hu (K.B.); somogyvari.ferenc@med.u-szeged.hu (F.S.); 2Department for Medical Microbiology and Hygiene, University of Mainz, D-55131 Mainz, Germany; friederike.loewenich@unimedizin-mainz.de

**Keywords:** *Bacteroides fragilis*, carbapenem resistance, imipenem, heterogeneous resistance, toxin-antitoxin pair, GNAT acetyltransferase toxin

## Abstract

Carbapenem-resistant *Bacteroides fragilis* strains usually emerge by an insertion sequence (IS) jump into the upstream region of the *cfiA* carbapenemase gene. However, intermediate or fully resistant *cfiA*-positive strains also exist. These do not have such IS element activations, but usually have heterogeneous resistance (HR) phenotypes, as detected by a disc diffusion or gradient tests. Heteroresistance is a serious antibiotic resistance problem, whose molecular mechanisms are not fully understood. We aim to characterize HR and investigate diagnostic issues in the set of *cfiA*-positive *B. fragilis* strains using phenotypic and molecular methods. Of the phenotypic methods used, the population analysis profile (PAP) and area under curve (AUC) measurements were the best prognostic markers for HR. PAP AUC, imipenem agar dilution and imipenemase production corresponded well with each other. We also identified a saturation curve parameter (quasi-PAP curves), which correlated well with these phenotypic traits, implying that HR is a stochastic process. The genes, on a previously defined ‘*cfiA* element’, act in a complex manner to produce the HR phenotype, including a lysine-acetylating toxin and a lysine-rich peptide. Furthermore, imipenem HR is triggered by imipenem. The two parameters that most correlate with the others are imipenemase production and ‘GNAT’ expression, which prompted us to suspect that carbapenem heteroresistance of the B. fragilis strains is stochastically regulated and is mediated by the altered imipenemase production.

## 1. Introduction

*Bacteroides fragilis* is the most significant anaerobic pathogen in terms of both its pathogenicity and our knowledge of its biology [1]. It resides in the normal intestinal microbiota of mammals, exerting important physiological activities there. It is also an opportunistic pathogen, being the most common species isolated from endogenous anaerobic infections such as intra-abdominal abscesses, pelvic abscesses, soft-tissue infections, wound infections and sepsis. Its parent genus, *Bacteroides,* with more than 60 described species, belongs to the Bacteroidetes phylum [1]. Regarding the number of antibiotic resistance mechanisms and resistance to various antimicrobials, members of the *Bacteroides* genus are the most antibiotic-resistant species among anaerobic pathogens. Of the pathogenicity factors of *B. fragilis*, its capsular polysaccharide (CPS) molecules, adhesins, oxygen tolerance and its enterotoxin are the most important [1]. The prevalence of *B. fragilis* in the intestinal microbiota varies person to person, but can range from 0% to 30% [2]. Its main interactions with hosts are mediated mainly by CPS types, which have immunomodulatory effects. This is important for physiologically shaping immune tolerance to *B. fragilis* in the intestine [3] and pathogenically in inducing abscesses [4]. Carbohydrate utilization is an additional characteristic common to *B. fragilis* and other *Bacteroides* species [5].

Antibiotic resistance can be categorized into the following prevalence classes: high (>60% for penicillins, cephalosporins and tetracyclines), moderate (10–60% for cephamycins, clindamycin, moxifloxacin and sometimes β-lactam/β-lactamase inhibitor combinations) and low (<5–10% for carbapenems, 5-nitroimidazoles and tigecycline) [2]. For carbapenems, the resistance rate is usually approximately 1–5% depending on the geographical location. It is mediated by a metallo-β-lactamase enzyme encoded by the *cfiA* gene. A well-known, but less well-characterized, property of *B. fragilis* strains is that they form two genetic divisions separable either by the presence of *cepA* penicillinase/cephalosporinase (Division I) or *cfiA* carbapenemase (Division II) genes, or by typing methods (genomic DNA homologies, ribotyping, multi-locus enzyme electrophoresis, multi-locus sequence typing, arbitrary primed polymerase chain reaction (AP-PCR) and matrix-assisted laser desorption ionization-time of flight mass spectrometry (MALDI-TOF MS)) [6]. In highly carbapenem-resistant *cfiA*-positive *B. fragilis* strains, the *cfiA* gene is usually activated by an insertion sequence (IS) element, which serves as a strong promoter for the expression of these resistance genes [6]. Several studies have already examined and described this configuration. However, the carbapenem resistance of *B. fragilis* is more complicated: (i) strains without an IS element have a silent *cfiA* gene and have low carbapenem M (<1 μg/mL), (ii) IS-activated *cfiA*-positive strains have high carbapenem minimal inhibitory concentrations (MICs, ≥16 μg/mL)) and (iii) *cfiA*-positive strains with an inactive IS element have elevated carbapenem MICs (>2 μg/mL) [7,8,9,10,11]. These latter strains also show heterogeneous carbapenem resistance phenotypes usually identified through Etest antimicrobial susceptibility test (AST) experiments [7,9,10,11].

Heterogeneous resistance (or hetero-resistance, HR) has been well examined in aerobic pathogens and some metronidazole-resistant *Clostridioides difficile* cases [12], but has been rarely reported on for other anaerobes. Although sometimes other phenomena are described as HR, true, ‘monoclonal’ or *bona fide* HR means that the subpopulations of cells with a higher resistance level exist in a bacterial culture or, equally, that the distribution of antibiotic susceptibilities of cells in the strain is wider than for normal bacterial populations [13]. HR is often noticed in routinely performed disc diffusion or gradient ASTs. However, the gold standard for detecting HR is recording its population analysis profile (PAP), whereby the proportion of surviving cells of a population is determined along with increasing antibiotic concentrations. PAP curves usually plot the logarithmic antibiotic concentration (x-axis) against the logarithm of the proportion of resistant cells (y-axis). HR is determined when the difference of MICs between the least and the most resistant population is ≥8 (≥three steps in two-fold dilutions) [13]. HR can also be observed in susceptible, intermediate (susceptible with increased exposure) and fully resistant categories of antibiotic resistance phenotypes [13]. As previously stated, HR for aerobic pathogens has been well described, with methicillin-resistant *Staphylococcus aureus* (MRSA) being the most well-known example [14]. Most ‘in the field’ MRSA strains exhibit HR, but the exact mechanism of HR in these and other aerobic species has not yet been well characterized [13]. For MRSA, experimental alterations in the expression of peptidoglycan synthesis genes and proteins often yielded HR phenotypes whose PAP profiles could be differentiated as (i) susceptible and homogeneous, (ii) some types of HR or (iii) highly resistant but also homogeneous [15]. The importance of HR and bacterial persistence is that it greatly increases the risk of developing full resistance during the use of antibiotics for ongoing infections [16].

The *cfiA* gene of *B. fragilis* also resides on a proposed genetic element (Figure 1), which, besides *cfiA*, codes for two acetyltransferase genes, ‘GNAT’ and ‘XAT’. The former shows homology to toxin–antitoxin (TA) system toxins [17]. TA gene pairs, with their toxic and labile neutralizing antitoxin activities, can kill post-segregational plasmid-less daughter cells (if harbored by plasmids) or halt cell division by stopping vital cellular functions (in the case of chromosomal localization). In this latter case, antibiotic persistence can emerge, meaning that during the replication stop, a small fraction of bacterial cells withstand the action of bactericidal antibiotics and, after their removal, the cells start dividing again [18].

Using some *B. fragilis* strains selected after carbapenem Etest ASTs, our aim is to prove the HR phenotype by using differently plotted survival curves. We also examine the phenotypic traits underlying these phenomena by performing calculations on optimal HR detection methods and study the gene expression of the ‘*cfiA* element’ to approximate how HR develops.

## 2. Results

### 2.1. Phenotypic Characterization and PAP Experiments

As our main aim was to further characterize imipenem HR, we chose the following test strains: three imipenem-susceptible *cfiA*-negative, nine *cfiA*-positive but ‘silent’ or heterogeneously imipenem-resistant in gradient tests and three imipenem-resistant, *cfiA*-positive, IS element-activated *B. fragilis* strains (Table 1). A typical imipenem gradient test example showing HR is displayed in Figure 2A, marking the start of the inhibition zone to the total disappearance of resistant colonies. The expression of imipenem HR phenotypes, as determined by gradient tests, varied from low to high (Table 1). We analyzed the HR behavior of colonies in the partial inhibition zone. For strains displaying low-grade HR phenotypes, the direct use of the cells in further gradient tests did not result in partial inhibition. However, if we allowed a ‘recovery’ period for those cells, incubating them on supplemented Columbia blood agar, the original HR phenotype recovered (without increased HR; Figure 2B–D), which demonstrated a monoclonal HR [14]. However, for strains showing a highly expressed HR phenotype, such as *B. fragilis* CZE60, some induction was observed as HR increased in these cases (data not shown).

### 2.2. PAP Curves, Assessment and Correlation of the Phenotypic Heteroresistance Parameters

In addition to PAP plots, plots without logarithmic axes (x- and y- axis direct—hyperbolic curves, x-axis logarithmic and y-axis direct—saturation curves) were also examined (Figure 3A–F), allowing more insight into the nature of HR. The saturation curves displayed some meaningful properties: it was interesting that, sometimes, at the lowest imipenem concentrations, we obtained fewer colony-forming units (CFU) than on the next higher-concentration plate (Figure 3B,E). We explained this by presuming that some dormant cells were present in the inoculating cell preparations (cultures suspended in PBS or BHIS) and that the higher, but non-selective, imipenem concentration, induced the cells to exit dormancy. The starting imipenem concentrations caused a smoothly decreasing curve, validating our sigmoid hypothesis, and the decrease in CFUs caused by increasing imipenem concentrations also tended to be somewhat continuous (Figure 3B,E). The saturation curves widened as the HR increased (Table 1, bS PBS and BHIS) and also produced the x_0_ value (Equation (1)), which was the inflection point or the maximum value of their derivative, the density function. In addition to these parameters, the PAP curves widened (the maximum being ≥3 times the minimum values, dil PBS and BHIS; Table 1).

To evaluate this curve’s widening in conjunction with other possible HR parameters, agar dilution MICs, HRI and imipenemase production were measured and compared for all strains (Table 1). We also recorded imipenem MICs on WC agar plates, since the latter was also used for PAP measurements. The PAP AUC ratios were also calculated, curves for which can be seen in Figure 4 and values in Table 1. All the phenotypic parameters are summarized and shown in Table 1. To analyze the data, the first variance analysis was performed according to HR categorization values (as PAP dilution increased by ≥3 or HRI was >0, Table 2). Almost all test parameters showed some potential for HR (Table 2). These parameters were also cross-correlated to see the connections between them, which was a cumulative assessment of relatedness (Table 3, where cells marked with different colors show differing degrees of correlation). Instead of examining the connection of only two parameters, it was a much more complete analysis. Through the cross-correlations, we also wanted to assess which had the best predictive and explanatory values for HR. We obtained a quite good rate of relatedness between the following parameters: (i) the x_0_ PBS and BHIS values correlated well with almost all the phenotypic parameters, (ii) the AUC PBS and BHIS and MIC Brucella and MIC WC also demonstrated a very high correlation and (iii) imipenemase production also correlated well with the other phenotypic parameters. This implied that (i) the AUC calculation best predicted HR, (ii) agar dilution predicted the resistance level quite well and (iii) an increased imipenemase production could be a cause of the HR phenotype.

### 2.3. Imipenem Induction of HR and Correlating the Molecular Characteristics of HR

A chromosomal segment (‘*cfiA* element’) containing the *cfiA* gene and a proposed TA gene pair with some insertional elements (MITE1, IS elements) were identified earlier as being characteristic of Division II *B. fragilis* strains (Figure 1). Additionally, during examination of the upstream regions of *cfiA* genes in *B. fragilis* strains, we identified a lysine-rich peptide (Lrp) in the ‘cfiA element’ (Figure 1). The ‘GNAT’ toxin gene showed a high homology to the elongation protein 3 (e = 1.03 × 10^−40^) [19] or to the AtaT-TacT-ItaT TA toxins (e = 3.4 × 10^−5^, classical members of acetylating toxin members of bacterial TA systems) [20] in protein BLAST conserved domain searches (Appendix A). Since a proposed TA gene pair resided on the ‘*cfiA* element’ and imipenem stirred the HR strain cells from dormancy, we attempted to induce or increase HR by imipenem treatment. The same phenotypic parameters for two strains (*B. fragilis* 3130i5 and CZE60i2) after serial imipenem inductions (five steps for 3130i5 and two steps for CZE60i2) were also recorded (Table 1). The detailed data for the *B. fragilis* 3130i5 HR induction process are shown in Figure 5. It can be seen that as imipenem concentration increased, HR, measured by E-tests and *cfiA* expression, ‘GNAT’ and ‘XAT’ genes and the ‘GNAT-XAT’ expression ratio, also showed increases. The increased HR in *B. fragilis* 3130i5 through induction with imipenem could also be clearly seen (Figure 3G), where the detected PAP curve widened and showed an increased resistance span. No growth was obtained when we exposed *B. fragilis* 3130 to a 128 mg/L imipenem concentration directly.

Induction experiments suggested that other ORFs on the ‘*cfiA* element’ (‘GNAT’, ‘XAT’ and Lrp) may also play a role in HR. Their contribution could be assessed based on their correlation with phenotypic traits. The cross-correlation of molecular traits also provided some insights into the possible action mechanism. The expression of ‘GNAT’ correlated with almost all the phenotypic traits, and imipenemase production correlated with the expressions of *cfiA* and ‘XAT’ (Table 3). ‘GNAT’ and ‘XAT’ showed significant correlation with each other (Table 3). *cfiA* gene expression showed a strong correlation with the composition of Lrp, as detected by the sequencing of its fragments by PCR amplification. Surprisingly, it also demonstrated a high degree of heterogeneity (Appendix A).

### 2.4. Time–Kill Curves

Since we hypothesized that ‘GNAT-XAT’ formed a TA pair that may also cause persister phenotypes, we performed some time–kill experiments to obtain further data (Figure 6). However, the curves were straight, reminiscent of antibiotic tolerance [21] even for the *B. fragilis* 638R control strain. Additionally, if a strain is tolerant to one antibiotic, it can also display tolerance to others through slow growth [21]. This was not the case for our strains, as shown in Appendix A. Nonetheless, we assumed that the tolerance to imipenem was mediated by other mechanisms in *B. fragilis* 638R and our other *cfiA*-positive test strains. This was reinforced, as imipenem induction did not cause alterations in the curves of imipenem-induced and non-induced strains (Figure 6).

**Table 1 antibiotics-11-00590-t001:** List of strains, recorded data on imipenem susceptibilities, PAP profiles, imipenemase production and molecular data.

*B. fragilis*	Ref.	Phenotypic Parameters ^a^												Molecular Parameters							
														*cfiA* and IS		qRT-PCR ^b^				Lrp ^c^	
		x_0_ PBS	x_0_ BHIS	bS PBS	bS HIS	d PBS	d BHIS	AUC PBS	AUC BHIS	MIC B	MIC WC	IP HRI	Ipase	*cfiA*	*cfiA*-IS	*cfiA*	‘GNAT’	‘XAT’	‘GNAT/XAT’	K no.	K%
Susceptible controls																					
NCTC 9343	-	−4.8142	−4.3392	−0.2092	−0.3902	2	2	1	1	0.125	0.064	0	0	-	n.a.	0	0	0	n.a.	n.a.	n.a.
638R	-	−4.4449	−4.9148	−0.2721	−1.0111	2	2	1.4757	1.4046	0.125	0.064	0	0	-	n.a.	0	0	0	n.a.	n.a.	n.a.
D39	[22]	−4.4350	−5.7965	−0.1012	−0.2192	2	2	1.4886	0.7642	0.5	0.064	0	0	-	n.a.	0	0	0	n.a.	n.a.	n.a.
Silent/HR ^d^																					
7979	This study	−3.0103	−3.5941	−0.4000	−0.8885	3	4	1.8045	1.5884	0.125	0.032	0	5.1	+	-	16.35	6.91	5.56	1.2428	7	26.9
3130	This study	−2.7685	−3.0168	−0.6269	−0.0451	3	6	1.8785	1.5532	8	8	0	2.6	+	-	1	1	1	1	5	13.5
3035	[7]	−2.8450	−2.7613	−0.3141	−0.3887	3	3	1.7810	1.4387	1	2	5	9.7	+	-	166.3	0.02	0.993	0.0201	9	25.0
SY69	[22]	−2.2774	−3.4300	−0.2062	−1.5269	5	6	2.0114	1.7008	4	4	5	10.1	+	-	92.0	17.7	32.6	0.5429	5	13.5
CZE65	[10]	−2.6235	0.9301	−0.9546	−1.1148	5	5	2.6027	2.3992	16	8	2	25.5	+	-	2.46	1.93	0.16	12.0625	7	26.9
CZE60	[10]	0.0886	−0.7637	−1.4586	−0.6320	3	4	2.7013	1.8329	16	16	5	33.3	+	-	12.91	7.78	0.49	15.8775	7	26.9
SLO8	[9]	1.8171	−4.0275	−0.6387	−1.1367	3	8	2.7294	2.2562	128	128	3	133	+	-	76.06	182.94	255.47	0.7161	8	25.0
HR-ind ^d^																					
3130i5	This study	−0.3129	−2.1373	−1.2807	−0.5161	4	8	2.9720	2.4650	4	4	5	28	+	-	5.91	52.18	18.67	2.80	5	13.5
CZE60i2	This study	3.1618	−2.7092	−0.4220	−2.0045	3	5	3.0608	2.4110	128	64	6	36.8	+	-	22.93	8.78	0.33	26.6061	7	26.9
With IS ^d^																					
De248514/19	This study	0.2977		−0.5716	−3.9655	3	4	2.4364	2.1085	16	64	3	134	+	IS*614*B	509.8	460.1	3.7	124.3514	8	25.0
1672	[8]		2.1815	−7.1023	−0.8173	2	3	2.6174	2.5047	256	256	4	189	+	IS*1168*	2.02	10.49	185.59	0.0565	7	26.9
TAL3636	[23]	3.5263	3.3190	−0.6621	−0.4304	2	4	2.8433	2.7366	128	256	0	354	+	IS*942*	97.5	0.024	0.697	0.0343	8	25.0

^a^ Abbreviations as follows: bS PBS—b parameter from PBS suspensions; bS BHIS—b parameter from BHIS broth cultures; d PBS—fold/dilution increase in PBS PAPs; d BHIS—fold/dilution increase in BHIS PAPs; Ipase—specific imipenemase production (u/mg cell extract). ^b^ Q-RT-PCR expressions and the relative expression (ratio of expressions) of ‘GNAT-XAT’. ^c^ K no.—number of lysines in Lrp; K %—the lysine content in Lrp in percentages. ^d^ Types of *cfiA*-positive strains: silent without IS elements but showing some HR, induced HR and strains with IS activated *cfiA* genes.

**Table 2 antibiotics-11-00590-t002:** Variance analysis of the examined traits of the test *B. fragilis* strains.

	Trait														
Grouping Category	x_0_ PBS	x_0_ BHIS	bS PBS ^a^	bS BHIS	d PBS	d BHIS	AUC PBS	AUC BHIS	MIC B	MIC WC	IP HRI	Ipase	*cfiA*	‘GNAT’	‘XAT’
Dilution change in PBS (*p*) ^b^	0.003	0.018	0.031	n.s. ^c^	**p.d. ^d^ (0.004)**	0.015	0.005	0.002	0.017	0.017	n.s.	0.007	0.033	0.029	0.031
Differences between groups ^e^	1–2, 1–3	1–3	1–3			1–2	1–2, 1–3	1–2, 1–3, 2–3	1–3	1–3		1–3	1–2	1–2	
Dilution change in BHIS (*p*)	0.01	0.03	0.021	n.s.	0.02	**p.d. (0.011)**	0.002	0.002	0.019	0.029	0.031	0.009	0.009	0.009	0.009
HRI (*p*)	0.001	n.s.	n.s.	n.s.	n.s.	n.s.	<0.001	<0.001	0.019	0.017	**p.d (<0.001)**	0.005	0.028	0.024	n.s
Differences between groups	1–2, 1–3						1–2, 1–3	1–2, 1–3			1–2, 2–3	1–3, 1–2		1–2	

^a^ Abbreviations are the same as Table 1. ^b^
*p*—significance values; ^c^ n.s.—non-significant; ^d^ Classifier traits (in bold, p.d.—per definition). ^e^ Numbers mean the following groups: 1—homogeneously susceptible; 2—heterogeneous resistance; 3—homogeneously resistant.

**Table 3 antibiotics-11-00590-t003:** Cross-correlation values of all strains of the traits examined (from Table 1).

	X_0_ BHIS	bS PBS	bS BHIS	d PBS	d BHIS	AUC PBS	AUC BHIS	MIC B	MIC WC	HRI	IPase	*cfiA*	’GNAT’	’XAT’	’GNAT/XAT’	n K	% K ^a^
X_0_ PBS	0.477	−0.613 ^b^	−0.477	0.29	0.594	0.916	0.873	0.932	0.924	0.519	0.96	0.669	0.695	0.457	0.227	0.256	0.366
	0.0809	0.0188	0.0809	0.301	0.0235	0.0000002	0.0000002	0.0000002	0.0000002	0.0537	0.0000002	0.00813	0.00507	0.0977	0.484	0.433	0.257
X_0_ BHIS		−0.725	0.125	0.177	0.281	0.625	0.721	0.578	0.523	0.345	0.538	0.143	0.108	0.183	−0.238	−0.081	−0.0931
		0.0018	0.648	0.514	0.3	0.0123	0.00196	0.0231	0.0429	0.199	0.0367	0.602	0.695	0.506	0.442	0.783	0.766
bS PBS			0.0679	−0.173	−0.396	−0.707	−0.779	−0.699	−0.685	−0.283	−0.699	−0.118	−0.47	−0.391	−0.126	0.11	0.0596
			0.802	0.532	0.138	0.00278	0.0000944	0.00326	0.00439	0.3	0.00326	0.667	0.0757	0.146	0.683	0.716	0.834
bS BHIS				−0.392	−0.327	−0.368	−0.382	−0.338	−0.308	−0.401	−0.437	−0.434	−0.652	−0.28	−0.545	−0.0663	−0.134
				0.142	0.23	0.171	0.154	0.209	0.257	0.134	0.101	0.104	0.00785	0.306	0.0623	0.834	0.667
d PBS					0.768	0.346	0.235	0.0807	0.00194	0.535	0.11	0.398	0.558	0.384	0.329	−0.487	−0.511
					0.0003	0.199	0.388	0.763	0.985	0.0382	0.686	0.138	0.0299	0.15	0.284	0.0998	0.0843
d BHIS						0.668	0.519	0.419	0.378	0.408	0.386	0.379	0.7	0.608	0.237	−0.528	−0.474
						0.00614	0.0463	0.117	0.158	0.127	0.15	0.158	0.00326	0.0158	0.442	0.0705	0.111
AUC PBS							0.9	0.804	0.759	0.57	0.824	0.437	0.663	0.43	0.294	−0.0663	0.0112
							0.0000002	0.0000002	0.00048	0.0252	0.0000002	0.101	0.00654	0.107	0.34	0.834	0.956
AUC BHIS								0.82	0.799	0.412	0.896	0.419	0.616	0.484	0.0559	0.0221	0.104
								0.0000002	0.0000002	0.124	0.0000002	0.117	0.0136	0.0662	0.852	0.939	0.733
MIC B									0.975	0.411	0.892	0.386	0.56	0.412	0.0497	0.247	0.329
									0.0000002	0.124	0.0000002	0.15	0.0287	0.124	0.869	0.429	0.284
MIC WC										0.341	0.908	0.422	0.543	0.412	−0.00352	0.319	0.413
										0.209	0.0000002	0.113	0.0353	0.124	0.974	0.295	0.173
HRI											0.443	0.477	0.591	0.369	0.172	−0.106	−0.166
											0.0946	0.0685	0.0192	0.171	0.572	0.733	0.588
IPase												0.626	0.68	0.532	0.035	0.434	0.521
												0.0123	0.00504	0.0397	0.904	0.15	0.0749
*cfiA*													0.547	0.493	−0.112	0.67	0.667
													0.0339	0.0597	0.716	0.0154	0.0169
’GNAT’														0.788	0.441	−0.155	−0.0745
														2 × 10^−7^	0.143	0.619	0.8
’XAT’															−0.364	−0.125	−0.0968
															0.233	0.683	0.749
’GNAT/XAT’																−0.214	−0.156
																0.484	0.619
n K																	0.988
																	0.0000002

^a^ Abbreviations are the same as Table 1. ^b^ Different colors were used depending on the strength of the correlations: blue–green—IrI ≥ 0.85 and *p* = 0.0000002; light green—IrI ≥ 0.7 or *p* < 0.01; yellow—IrI ≥ 0.5, or *p* < 0.05. Boxes marked with A–E indicate common relations and are discussed in the text.

## 3. Discussion

This study revealed that carbapenem heteroresistance is a characteristic phenotype of some *cfiA*-positive *B. fragilis* strains. The phenotypic parameters studied as the saturation curve x0 and b, the PAP curve dilution and AUC, the imipenem MIC and HRI values and the specific imipenemase activities of the strains were interconnected and predicted HR well. The most important and central parameter was the imipenemase production, which could mediate the resistance and affect the other phenotypic parameters as it was proportional to the other parameters observed: agar dilution MICs, PAP AUC ratios and the saturation curve (simpler PAP curve) extension in PAPs of agar plate-grown cells. In our opinion, all the studied parameters could predict HR, but PAP AUC was the best. However, since most of these parameters were continuous in form from low to high imipenem MICs and HR parameters, continuing to use PAP curve extensions could also be regarded as a very good method. The PAP AUC method was also suggested as a good prediction parameter for the reduced glycopeptide susceptibility of staphylococci [24].

We related the HR phenotype to stochastic processes. According to our hypothesis, this was due to the action of a proposed toxin (‘GNAT’) that may stop growth, but also allows viability under antibiotic-exposed circumstances. The primary finding supporting this was the widening of the PAP saturation curve parameter, b, which could be regarded as a standard deviation parameter. To improve the discussion of HR and persistence, Brauner and Balaban suggested the term heterotolerance for HR [25]. In more thoroughly investigated tolerance and persistence mechanisms, it has also been suggested that wider distributions, and those above a certain persistence factor threshold, yield more persisters [24]. We believe this is true for the carbapenem heteroresistance of *B. fragilis*, as we also detected a widening of our saturation curves. Imipenemase activity values, as an effector mechanism, also correlated with most of the phenotypic parameters of HR [26].

We propose an HR mechanism of *cfiA*-positive *B. fragilis* strains as follows: (i) *cfiA* is expressed proportionally to HR, and (ii) the parallel expression of ‘GNAT-XAT’ allows reduced cellular activities. We conclude the same from experiments in which we induced imipenem HR by imipenenem; however, to obtain a more detailed picture about this, experiments are under way in our laboratory to determine the promoters of *cfiA* and ‘GNAT-XAT’, how they act individually and in conjunction with other promoters, what is the biochemical nature of ‘GNAT’ and ‘XAT’, do they form a TA pair and what is the role of the lysine-rich peptide in the ‘*cfiA* element’. It is conceivable through the above that ‘GNAT’ acts through the acetylation of a lysine in a ribosomal protein or in tRNA^Lys^ molecules and Lrp may modify these actions.

At present, no particular common mechanism was found to explain HR in other bacteria. However, in some cases, regulatory proteins were involved as well [27,28,29,30,31], which we believe may also produce stochastic regulation. Additionally, monoclonal heteroresistance could also emerge by the tandem duplication of DNA segments of the effector genes of *Acinetobacter baumannii* [32], *Escherichia coli* [33], *Klebsiella pneumoniae* [34], *Pseudomonas aeruginosa* [35], *Salmonella typhimurium* [36] or *Streptococcus pneumoniae* [37]. However, this latter mechanism can result in a variable number of repeats in the cells of a given population, which can be both stochastic and difficult to detect. Recently, for the aminoglycoside HR of *A. baumannii* in a recA-negative background, a modest copy number variation of the *aadB* gene-containing integron was linked to HR. The step causing the copy number increase was hypothesized as a stochastic process [32]. In earlier experiments, we did not observe that *cfiA* or the ‘*cfiA* element’ had copy number variations or that the *cfiA* promoter was invertible (data not shown), as CPS promoters usually are in *B. fragilis*. The role of global regulatory systems ((p)ppGpp, *relA*, *spoT*) in bacterial persistence was proven, something which we would like to examine regarding the HR of *B. fragilis* [26].

For TA systems, the prominent role of governing persistence was attributed, but some parallels between HR and persister phenotypes could also be drawn: HR can be regarded as concentration-dependent, while persistence can be regarded as a time-dependent survival phenomenon. In our opinion, the stochastic hypothesis of carbapenem HR of *B. fragilis* may facilitate research into this being the case in other HR systems as well.

In this study, we analyzed extensively the phenotypic parameters of control and HR *B. fragilis* strains, which yielded scattered but statistically evaluable data enabling novel description by saturation curves. In summary, these investigations into the various HR traits revealed that while the AUC PAP method was the best predictor/classifier for HR, other traits could also be considered suitable. Among the phenotypic traits examined, the saturation curve, PAP AUC, agar dilution and imipenemase activities correlated well. This indicated that they were also good predictors and were linked to the HR mechanism, for which imipenemase production could be the primary contributor. The calculation of the widening of the saturation and PAP curves, for the PBS suspensions, showed a good correlation with the PAP AUCs, agar dilution MICs and imipenemase production. Therefore, we saw our stochastic explanation of the nature of HR as compelling. This latter point was also supported by the fact that imipenem HR could be induced by imipenem and that the expression of the ‘*cfiA* element’ genes (‘GNAT’, ‘XAT’, and *cfiA*) correlated. ‘GNAT’ can act as a toxin causing dormancy, and these genes may form a complex interaction.

## 4. Materials and Methods

### 4.1. Bacterial Strains and Cultivation

The test strains used in this study are listed in Table 1. The strains were selected from our collection stored at −70 °C in brain–heart infusion broth (BHI) containing 20% glycerol. Their cultivation was performed on anaerobic Columbia blood agar plates (Columbia agar supplemented with 2.5% defibrinated sheep blood, 1.25% laked sheep blood, 300 mg/L L-cysteine and 1 mg/L vitamin K_1_), on Wilkins–Chalgren (WC) agar or in supplemented BHI broth (BHIS, with the addition of 0.5% yeast extract, 5 mg/L hemin and 1 mg/L vitamin K_1_) at 37 °C under anaerobiosis (85% N_2_, 10% H_2_ and 5% CO_2_) in an anaerobic cabinet (Concept400, Ruskinn, UK).

### 4.2. MIC Determinations, Recording of Population Analysis Profiles and Time–Kill Curves

Gradient tests (E-test, bioMérieux, France) were conducted on supplemented Columbia blood agar plates. Agar dilution was carried out as recommended by CLSI [38] on supplemented Columbia blood agar plates. Agar dilution was also performed on WC plates as the PAP records were also determined on this media.

In PAP experiments, we used the following cell suspensions/cultures: (1) 0.5 McFarland phosphate-buffered saline (PBS, 137 mM NaCl/2.7 mM KCl/1.8 mM KH_2_PO_4_/10 Na_2_HPO_4_ pH 7.2) cell suspensions taken after 48 h cultivations on Columbia blood agar plates or (2) overnight incubated BHIS broth culture. Optical density (at 600 nm) was measured by spectrophotometer (Thermo Scientific, Budapest, Hungary) for later normalization. Ten-fold dilutions were composed in PBS and 100 μL inocula were spread on Wilkins-Chalgren agar plates with an appropriate concentration of imipenem (from the 0.008–1024 μg/mL range), which was determined by trials to yield 50–500 countable colonies per plate. Two independent experiments (biological replicates) were carried out using two–three parallels (technical replicates) for each concentration. The inoculated WC plates were then incubated anaerobically for 48 h and, afterwards, colony counts were determined by a gel documentation system (PXi, SYNGENE, Oxford, UK).

Time–kill curves were recorded by plating after 0 h, 2 h, 4 h, 10 h and 24 h of incubation on antibiotic-free WC agar plates, 100 μL aliquots of serial 10-fold dilutions of PBS suspensions with a turbidity of 0.5 McFarland. These also contained 32-fold higher imipenem concentrations than the original imipenem MICs. The WC plates were incubated in anaerobiosis for 48 h. Colonies were counted as described above.

### 4.3. Imipenemase Activity Measurement and Induction of HR by Imipenem Treatment

In total, 8 mL of overnight BHIS cultures was centrifuged (at 4 °C, 8000 rpm, 10 min), washed 3 times with cold PBS and sonicated. The crude cell extracts were then used for imipenemase activity determination in 1 mL UV-transparent plastic cuvettes in an Assay buffer (50 mM HEPES, 25 μM ZnSO_4_, pH 7) using 0.1 mM imipenem concentration and an adjusted enzyme volume to obtain a linear decay of imipenem followed at 299 nm. The results were expressed by 1 pmole imipenem hydrolyzed per 1 min (U) and standardized by the protein content of the extracts (U/mg). Protein concentrations were determined by the Qubit Protein Assay Kit (Thermo Fisher Scientific, Budapest, Hungary).

Imipenem and imipenem heteroresistance were increased through 10 mL anaerobic BHIS cultures of *B. fragilis* 3130 and CZE60 being exposed to stepwise increments (0, 2, 8, 32 and 128 μg/mL for *B. fragilis* 33130 and 0, 32 and 128 μg/mL for *B. fragilis* CZE60) of imipenem concentrations. This was achieved by subculturing the lower imipenem concentration, containing stationary phase cultures, to the next level of imipenem-concentrated BHIS broth, to obtain an OD_600_ of 0.05–0.1. We let it propagate to a stationary phase (OD_600_ of 0.7–1.5) which took more time, from 1 to 4 days, as the imipenem concentrations increased.

### 4.4. Conventional PCR, Nucleotide Sequencing and qRT-PCR Experiments

Conventional PCRs and the nucleotide sequencing of some of its PCR products were carried out as described previously. PCR primer sequences and cycling conditions are contained in Appendix A.

To examine the ‘*cfiA* element’ constant gene (‘GNAT’, ‘XAT’ and *cfiA*) expression levels, total RNA was isolated (HighPure RNA Isolation Kit, Roche) from the *cfiA*-positive test strains and we performed subsequent qRT-PCR in an RT-PCR instrument (StepOne, Life Technologies). The 10 μL final volume PCR reactions contained 5 μL SYBR Green mastermix (Verso 1-Step RT-PCR Mastermix with ROX, Thermo Fisher Scientific, Budapest, Hungary), 0.7 μM primers and 1 μL RNA sample.

### 4.5. Curve Plotting, Curve Parameter Calculation, Statistical Evaluation and Bioinformatics

Means and standard deviations were calculated after normalization of the OD_600_ values in MS Excel. The highest colony counts for each type of measurement were then regarded as 1, and smaller colony counts were expressed as a fraction of that. The values of growth fraction for each imipenem concentration obtained this way were then plotted (Sigmaplot 12) by direct axes (quasi hyperbolic curves), logarithmic x-axis of imipenem concentrations and direct y-axis of growth fraction (‘saturation curves’) and with both axes logarithmic (classical PAP curves). For the saturation curves, the following Equation (1) (3-parameter sigmoid models) was used to assess the slope (b parameters) of the HR growth:y = a/(1 + exp(−(x − x_0_)/b))(1)

In estimating HR by calculating the ≥3 dilution decreases (dilution change) in the PAP curves, we started to count once the difference in the number of colonies was in the ten-fold range (since there were minimal differences in colony counts in the low-concentration ranges). For PAP curves, the extensions in imipenem concentrations and area under curve (AUC) ratios were calculated after all cell content was normalized to 10^10^ CFU. PAP AUCs were calculated by Sigmaplot 12 and divided by the value of *B. fragilis* NCTC 9343 PAPs (from PBS suspensions and BHIS cultures). We also included the HR index of gradient tests (HRI), which expressed the number of step differences in 2-fold increments between the full growth and full inhibition values (Table 1).

To estimate the congruence between the test strains various phenotypic and molecular parameters, 1-way ANOVA with different HR grouping parameters ((i) ≥ 3-fold changes in PBS PAP dilutions, (ii) ≥ 3-fold changes in BHIS PAP dilutions and (iii) HRI>0) was used. The Holm–Sidak method (normal distributions) or Dunn’s methods were used (Sigmaplot 12) between group differentiations. In addition, Spearman rank correlation calculations were performed to estimate congruences between recorded parameters (Sigmaplot 12).

Alignments of nucleotide and amino acid sequences were performed by Lasergene 17 (DNAStar Inc., Madison, WI, USA) using the Clustal Ω algorithm.

## Figures and Tables

**Figure 1 antibiotics-11-00590-f001:**
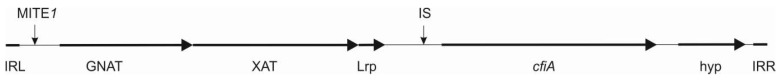
An updated scheme of the *cfiA* ’element’. IRL and IRR inverted repeats bordering the element; ‘GNAT’—ORF coding a Gcn5-like acetylase protein; ‘XAT’—another acetylase protein; Lrp—a short ORF coding for a lysine-rich peptide; hyp—an ORF coding for a hypothetical protein; MITE*1* with arrow—possible insertion of miniature transposable element 1; IS with arrow—possible insertion of a *cfiA*-activating IS element.

**Figure 2 antibiotics-11-00590-f002:**
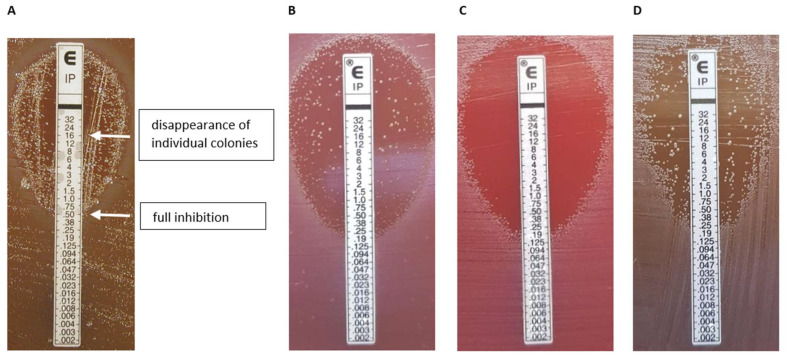
Etest HR phenotypes. A typical Etest phenotype (*B. fragilis* 16997) [8] showing the reading annotation; in this example 0.5–(16), the first value expresses the inhibition of full growth, then after a dash, the value indicates in parenthesis where the inside colonies disappeared. (**A**). The ‘inheritance’ of the Etest HR phenotype (**B**–**D**). Original appearance (**B**); an inner colony direct Etest (**C**); an Etest result of an inner colony after an intermediate cultivation on a Columbia blood agar (**D**).

**Figure 3 antibiotics-11-00590-f003:**
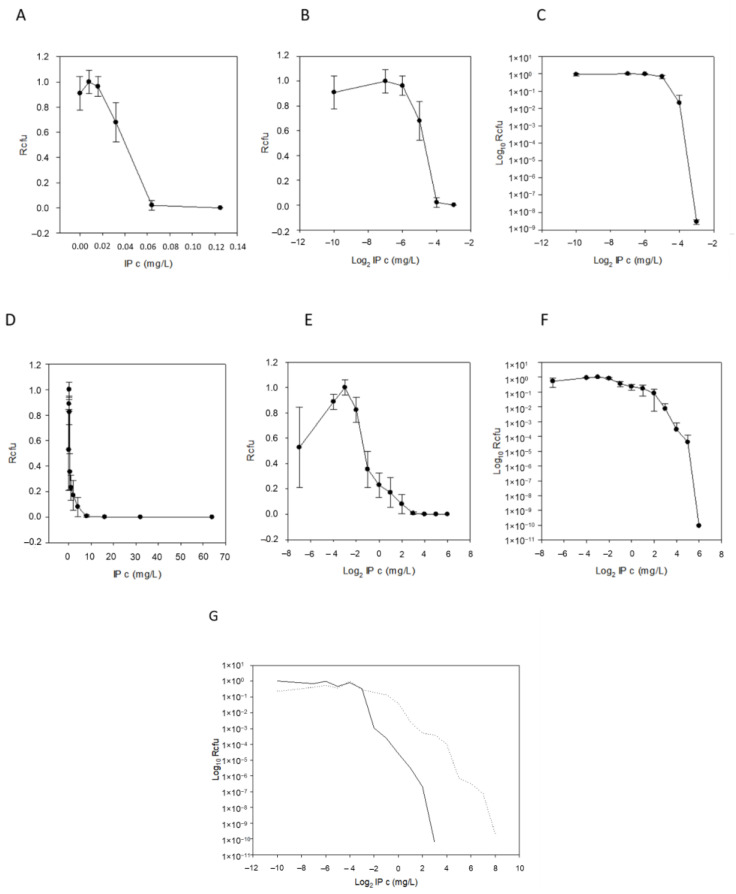
Demonstration of PAP curves. Imipenem concentration vs. growing colony number plots with different scale axes and with error bars for the imipenem susceptible and *cfiA*-negative *B. fragilis* NCTC 9342 (**A**–**C**) and for HR and *cfiA*-positive *B. fragilis* CZE60 (**D**–**F**). Direct plots (hyperbola-like curves, (**A**,**D**)), logarithmic IP concentration (saturation-like curves, (**B**,**E**)) and both axes logarithmic (classic PAP curves, (**C**,**F**)). (To be able to logarithmize zero concentration of imipenem, we used an 8-fold smaller value, such as 0.004 mg/L, instead of 0.032 mg/L). Comparison of the PAP curves of imipenem uninduced (solid line) and imipenem-induced (dashed line) *B. fragilis* 3130 (**G**).

**Figure 4 antibiotics-11-00590-f004:**
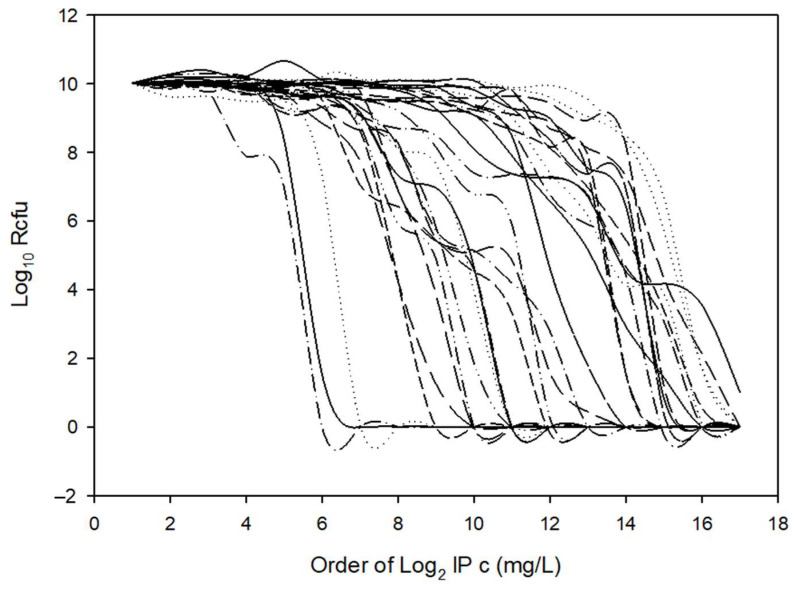
Cumulative PAP plots used in AUC calculations of our 15 *B. fragilis* test strains (Table 1) cultured on solid (supplemented Columbia blood agar) or in liquid (BHIS) media.

**Figure 5 antibiotics-11-00590-f005:**
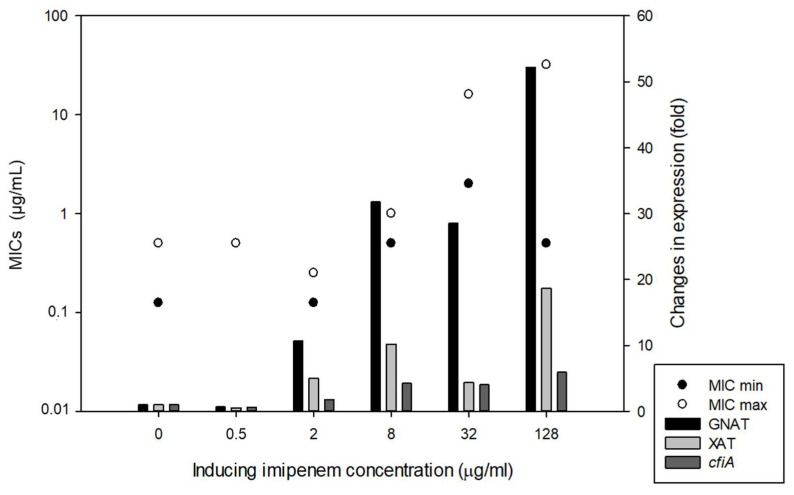
Induction of HR in *B. fragilis* 3130 with increasing imipenem concentration in the culture media (BHIS). The left-hand y-axis denotes the Etest AST values (full and partial inhibition marked as minimal and maximal MICs) and right-hand y-axis shows the gene expression changes for the strains in the induction experiments, respectively.

**Figure 6 antibiotics-11-00590-f006:**
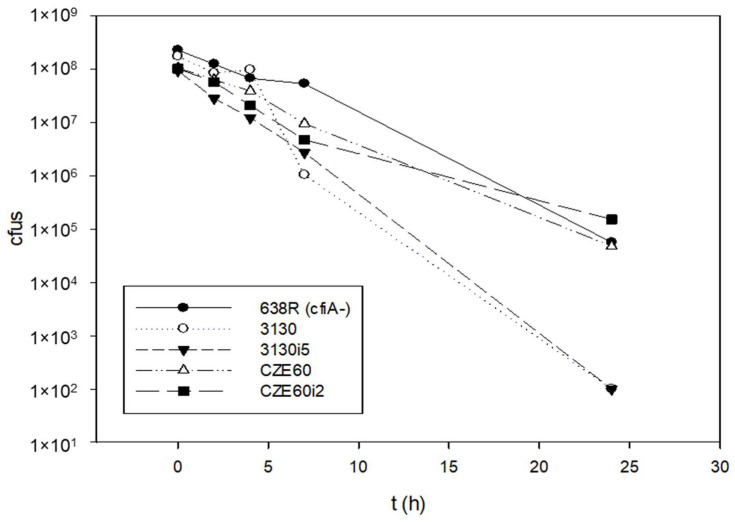
Time–kill curves of *B. fragilis* strains with or without *cfiA* gene. *B. fragilis* 638R was *cfiA*-negative, all the other strains were *cfiA*-positive. The imipenem non-induced and induced variants pf *B. fragilis* 3130 and CZE60 did not show much difference in their killing curves.

## Data Availability

Not applicable.

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
