# Peer review of "Phenotypic and Molecular Characterization of Carbapenem-Heteroresistant Bacteroides fragilis Strains"

_antibiotics, 2022, doi:10.3390/antibiotics11050590_

Round 1
Reviewer 1 Report
General comment: This study highlights an interesting topic! The authors have done a good job, but the referee has some comments on improving the work.
- In line 12, find another word for "insertions" because it's after IS and doesn't sound good in English.
- Throughout the entire manuscript, it is noted that references are superscript, but must be placed in brackets, for example [1], and so on.
- The words AP-PCR, MALDI-TOF MS, in lines 54,55, need their long names first and then their acronyms.
- In line 61, put MIC's long name first and then its acronym, because it's used here for the first time.
- The word "cases" in line 66 is not required to be expressed in italics.
- In line 67, put HR's long name first and then its acronym.
- Regarding the aim, please find another verb instead of “examined” and it would be better to review the entire sentence (lines 98-100).
- In the Results section, please insert Table 1 when it is first cited (line 113), the same for Table 2 (line 160) and Table 3 (line 162). This section needs to be corrected, revised after changes. In this way it would be clearer to better understand the work done.
- Put the acronym of A. baumannii in the second time used.
- In the Materials and Methods section, enter the city and state of Thermo Scientific used for the first time, of the Qubit Protein Assay Kit, and the state of Madison ... Also, in this section, improve the sentences where it started with a number, such as (10-fold) and (8 mL).
- As a final comment, the English of the manuscript should be revised after corrections made.
Author Response
Answers to Referee 1
General comment: This study highlights an interesting topic! The authors have done a good job, but the referee has some comments on improving the work.
We thank the examination of our manuscript and also the helpful comments.
- In line 12, find another word for "insertions" because it's after IS and doesn't sound good in English.
Answer: Corrected.
- Throughout the entire manuscript, it is noted that references are superscript, but must be placed in brackets, for example [1], and so on.
Answer: This was retained since the template file in which the necessar changes should be made contained this format. The final copy editing should correct these.
- The words AP-PCR, MALDI-TOF MS, in lines 54,55, need their long names first and then their acronyms.
Answer: Corrected.
- In line 61, put MIC's long name first and then its acronym, because it's used here for the first time.
Answer: Done
- The word "cases" in line 66 is not required to be expressed in italics.
Answer: Corrected.
- In line 67, put HR's long name first and then its acronym.
Answer: Done.
- Regarding the aim, please find another verb instead of “examined” and it would be better to review the entire sentence (lines 98-100).
Answer: Done.
- In the Results section, please insert Table 1 when it is first cited (line 113), the same for Table 2 (line 160) and Table 3 (line 162). This section needs to be corrected, revised after changes. In this way it would be clearer to better understand the work done.
Answer: Done.
- Put the acronym of A. baumannii in the second time used.
Answer: Done.
- In the Materials and Methods section, enter the city and state of Thermo Scientific used for the first time, of the Qubit Protein Assay Kit, and the state of Madison ... Also, in this section, improve the sentences where it started with a number, such as (10-fold) and (8 mL).
Answer: Done.
- As a final comment, the English of the manuscript should be revised after corrections made.
Answer: Before submission we have corrected our manuscript by a native Enlish and this revision should show the corrections made by the authors therefor we will utilize the grammar correction service of the Publisher.
Reviewer 2 Report
Most ‘in the field’ MRSA strains exhibit 80
HR, but the exact mechanism of HR in these and other aerobic species has not yet been 81
well characterized 13. For MRSA, experimental alterations in the expression of peptidogly- 82
can synthesis genes and proteins often yielded HR phenotypes whose PAP profiles can 83
be differentiated as (i) susceptible and homogeneous, (ii) some types of HR or (iii) highly 84
resistant but also homogeneous 16. The importance of HR and bacterial persistence is that 85
it greatly increases the risk of developing full resistance during the use of antibiotics for 86
ongoing infections 14- The sequence of reference is not in ascending order.
Figure 2. Etest HR phenotypes. A typical Etest phenotype (B. fragilis 16997)8 showing the read- 125
ing annotation, e.g. 1-(16), full inhibition value first, then after a dash, the disappearance of the in- 126
side colonies in parenthesis (A). The ‘inheritance’ of the Etest HR phenotype (B-D). Original appear- 127
ance (B), an inner colony direct Etest (C), an Etest result of an inner colony after an intermediate 128
cultivation on a Columbia blood agar (D).- Legend needs to be elaborated
Table 1. List of strains, recorded data on imipenem susceptibilities, PAP profiles, imipenemase production and molecular data- More references are required in table 1.
Table 3 needs to explain more in the manuscript.
The discussion section requires more description of the findings.
Author Response
Answers to Referee 2
We thank the thorough examination of our manuscript and the helpful comments.
Most ‘in the field’ MRSA strains exhibit 80
HR, but the exact mechanism of HR in these and other aerobic species has not yet been 81
well characterized 13. For MRSA, experimental alterations in the expression of peptidogly- 82
can synthesis genes and proteins often yielded HR phenotypes whose PAP profiles can 83
be differentiated as (i) susceptible and homogeneous, (ii) some types of HR or (iii) highly 84
resistant but also homogeneous 16. The importance of HR and bacterial persistence is that 85
it greatly increases the risk of developing full resistance during the use of antibiotics for 86
ongoing infections 14- The sequence of reference is not in ascending order.
Answer: Done.
Figure 2. Etest HR phenotypes. A typical Etest phenotype (B. fragilis 16997)8 showing the read- 125
ing annotation, e.g. 1-(16), full inhibition value first, then after a dash, the disappearance of the in- 126
side colonies in parenthesis (A). The ‘inheritance’ of the Etest HR phenotype (B-D). Original appear- 127
ance (B), an inner colony direct Etest (C), an Etest result of an inner colony after an intermediate 128
cultivation on a Columbia blood agar (D).- Legend needs to be elaborated
Answer: Done.
Table 1. List of strains, recorded data on imipenem susceptibilities, PAP profiles, imipenemase production and molecular data- More references are required in table 1.
Answer: Done.
Table 3 needs to explain more in the manuscript.
Answer: We wrote more about Table 3.
The discussion section requires more description of the findings.
Answer: We complemented the Discussion.
Reviewer 3 Report
The scientific article written by Baaity et al. addresses the carbapenem heteroresistance in B. fragilis using phenotypic and molecular methods. The study is interesting as there are not much research dealing with the topic of heterogenous resistance in anaerobic pathogens. Therefore, the significance of this study lies in the fact that heteroresistance in such pathogens has not been frequently explored earlier and the methods are not well established. I find the experiments well designed and executed. However, there are some caveats to the data presentation. I have the following comments to improve the manuscript:
- For phenotypic characterization the authors used population analysis profile (PAP) and area under curve (AUC) methods that are the preferred approaches to address heterogenous resistance. I am not sure about the motivation behind the methods used for the molecular characterization. Can the author please explain Why they did not opt for other approaches to investigate the role of “cfia element” such as knockout approach etc? In my opinion that would have been a nice control strain to have for phenotypic and molecular characterization.
- Also, the authors clearly mention about what method they used for phenotypic characterization and conclusion in the abstract but I missed the molecular part again in abstract so I could not distinguish between transition from phenotypic to molecular approaches as they have not discussed much about it in the text also (mainly in tables).
- Figure 4: Which 15 B. fragilis strains. Are these the replicate of same strains or the different strains? If different add legends. It is not clear what message the authors want to convey.
- Introduction section: Add references if you are making a statement that is not based on this study. Multiple references are missing. For example, Line 29 “the most significant” add reference. Is it the most significant or one of the important anaerobic pathogens?
- Discussion section: What is “earlier unpublished data” Why not to include any unpublished data that authors used to make a conclusion in supplementary instead writing data not shown (Line 123).
- In my opinion writing style and presentation could be improved. There are too many tables with lot of details that is difficult to follow up and reader have to make own notes to keep track of details. Would it be possible to have these as graphs? If not, please check the headings of the columns, so they are reduced. More info can be added in the description of the table instead.
- There is no reference for the strain NCTC 9343 and 638R in the table 1? Is it on purpose? For all other strains I can see references.
- Some statements are long/connected by “and” or “brackets”. I will recommend splitting such sentences into two to make it easier for reader to understand. For example, M&M “In PAP experiments, the starting culture were either 0.5 McFarland phosphate-buffered saline…….”
Author Response
Referee 3
The scientific article written by Baaity et al. addresses the carbapenem heteroresistance in B. fragilis using phenotypic and molecular methods. The study is interesting as there are not much research dealing with the topic of heterogenous resistance in anaerobic pathogens. Therefore, the significance of this study lies in the fact that heteroresistance in such pathogens has not been frequently explored earlier and the methods are not well established. I find the experiments well designed and executed. However, there are some caveats to the data presentation. I have the following comments to improve the manuscript:
We thank the thorough examination of our manuscript and the improving comments.
- For phenotypic characterization the authors used population analysis profile (PAP) and area under curve (AUC) methods that are the preferred approaches to address heterogenous resistance. I am not sure about the motivation behind the methods used for the molecular characterization. Can the author please explain Why they did not opt for other approaches to investigate the role of “cfia element” such as knockout approach etc? In my opinion that would have been a nice control strain to have for phenotypic and molecular characterization.
Answer: We thank this suggestion which in part is ongoing in our laboratory (cloning of the genes and studying the effects of the element) or is planned to carry out in the future (knock-out). However the present study also requested a significant amount of work from us.
- Also, the authors clearly mention about what method they used for phenotypic characterization and conclusion in the abstract but I missed the molecular part again in abstract so I could not distinguish between transition from phenotypic to molecular approaches as they have not discussed much about it in the text also (mainly in tables).
Answer: We completed the Abstract with discussion of the molecular data. In the text and in the succession of the figures/tables the induction experiment is making the link. This originated from our earlier experiments when we did imipenem inductions too, got interesting findings with that and which may lead us to map better the interactions of the genes of the ’cfiA element’ too.
- Figure 4: Which 15 B. fragilis strains. Are these the replicate of same strains or the different strains? If different add legends. It is not clear what message the authors want to convey.
Answer: We completed the title to be more informative.
- Introduction section: Add references if you are making a statement that is not based on this study. Multiple references are missing. For example, Line 29 “the most significant” add reference. Is it the most significant or one of the important anaerobic pathogens?
Answer: We added a reference.
- Discussion section: What is “earlier unpublished data” Why not to include any unpublished data that authors used to make a conclusion in supplementary instead writing data not shown (Line 123).
Answer: We corrected this issue by writing Data not shown.
- In my opinion writing style and presentation could be improved. There are too many tables with lot of details that is difficult to follow up and reader have to make own notes to keep track of details. Would it be possible to have these as graphs? If not, please check the headings of the columns, so they are reduced. More info can be added in the description of the table instead
Answer: We modified the titles of some tables, however, if we wanted to present all of our gathered data it required this number of figures and tables. We also completed the titles.
.
- There is no reference for the strain NCTC 9343 and 638R in the table 1? Is it on purpose? For all other strains I can see references.
Answer: The B. fragilis NCTC 9343 and 638R strains are well known for the community working with anaerobes and can be find in culture collections too. So this is why we did not regard to reference them necessarily.
- Some statements are long/connected by “and” or “brackets”. I will recommend splitting such sentences into two to make it easier for reader to understand. For example, M&M “In PAP experiments, the starting culture were either 0.5 McFarland phosphate-buffered saline…….”
Answer: We corrected this sentence.